# Global and regional burden and inequalities of oral conditions in children, adolescents, and young adults (0–39 years), 1990 to 2021

Yue Chen[1,2☯], Zhishen Jiang[1,3☯], Liu Liu [1]*, Jian Pan[1,3], Yubin Cao[1,3,4], Wenli Lai[1,2], Hu Long[1,2]*

1 State Key Laboratory of Oral Diseases & National Centre for Stomatology & National Clinical Research Centre for Oral Diseases, West China Hospital of Stomatology, Sichuan University, Chengdu, China, 2 Department of Orthodontics, West China Hospital of Stomatology, Sichuan University, Chengdu, China, 3 Department of Oral and Maxillofacial Surgery, West China Hospital of Stomatology, Sichuan University, Chengdu, China, 4 Department of Evidence-Based Stomatology, West China Hospital of Stomatology, Sichuan University, Chengdu, China

☯ These authors contributed equally to this work.

* liul@scu.edu.cn (LL); hlong@scu.edu.cn (HL)

## Abstract

Previous Global Burden of Disease studies often encompassed the entire age spectrum or treated adolescents and young adults as a single cohort. This methodology has limited the opportunity for a detailed analysis and modelling of oral health within specific subgroups of the younger population. This study examines the lifecycle-specific burden, trends, and inequalities of oral disorders among children, adolescents, and young adults from 1990 to 2021, utilizing data from the Global Burden of Disease 2021. Age-standardized Years Lived with Disability (YLD) rates were calculated by sex, age subgroups, and sociodemographic index (SDI) quintiles, followed by incidence and prevalence calculations. The study utilized Joinpoint regression, decomposition analysis, slope/concentration index, and sociodemographic attribution analysis to assess the epidemiology of oral disorders. As a result, in 2021, there were 6.22 (3.40-10.57) million YLDs associated with oral disorders among children, adolescents, and young adults globally. Temporal trends indicated a post-pandemic rise in deciduous caries among children. The number of YLDs of caries in children, adolescents, and young adults has all increased despite declining age-standardized YLD rates in 1990–2021 due to population growth in lower-SDI locations. The number and age-standardized YLD rates of periodontal disease and edentulism have risen among adolescents and young adults, exacerbated by worsened epidemiology and population growth. Despite improvements in 2021 compared to 1990, inequalities in periodontal disease burdens persist, disproportionately affecting lower SDI locations. Findings of this study reveal that oral disorder epidemiological metrics among individuals aged 0–39 have generally improved for caries but worsened for periodontal diseases and edentulism. Population growth in lower-SDI locations primarily drove the

**Data availability statement:** Data sources and codes used in the Global Burden of Disease Study 2021 are publicly available at http://ghdx.healthdata.org/gbd-results-tool. The map of global national administrative boundaries used in this work is from the Resource and Environment Science and Data Platform, Institute of Geographic Sciences and Natural Resources Research, Chinese Academy of Sciences, and is publicly available at https://www.resdc.cn/data.aspx?DATAID=205.

**Funding:** This work was supported by the National Natural Science Foundation of China (Grant 82471019 to H.L., Grant 82301106 to Y.C., and Grant 82471022 to W.L.), and the Natural Science Foundation of Sichuan Province (Grant 2024NSFSC1585 to Y.C.). The funders had no role in study design, data collection and analysis, decision to publish, or preparation of the manuscript.

**Competing interests:** The authors have declared that no competing interests exist.

global increased burdens. SDI-related inequalities disproportionately concentrated periodontal disease burdens in lower-SDI locations. Targeted healthcare resource allocation is essential for youth population to address the increased burdens and inequalities and enhance universal health coverage.

## 1. Introduction

The Global Burden of Disease (GBD) 2021 framework is a critical resource for health estimates for various stakeholders, including patients, clinicians, researchers, and policymakers [1]. Oral disorders, including untreated caries in deciduous and permanent teeth, periodontal diseases, and edentulism, constitute a significant yet underprioritized aspect of the global noncommunicable disease burden. While previous GBD studies have integrated oral disorders with noncommunicable disease strategies and highlighted them as public health challenges contributing to health inequalities [2–5], gaps remain in understanding the lifecycle-specific epidemiology of these conditions, particularly among children, adolescents, and young adults aged 0–39 years (young populations).

Existing research has established some traditional concepts regarding the distribution of oral disorder burdens, influencing healthcare and research resource allocation. Caries predominantly affects young individuals [2,4], leading to a focus of resources on children, while periodontal diseases and edentulism are typically associated with late adulthood [6–9]. However, recent evidence suggests shifting disease trajectories, including delayed caries progression in permanent teeth and earlier onset of periodontal diseases and edentulism among adolescents and young adults [6,10,11]. These changes may be unevenly distributed globally, reflecting socioeconomic disparities and highlighting the need for a comprehensive analysis of young populations aged 0–39 years.

Previous GBD studies have provided valuable insights into the global, regional, and national burden of oral disorders among populations, with a particular focus on those aged 10–24 and without age limitations [2–4,6,12,13]. However, gaps remain in understanding the lifecycle-specific epidemiology and inequalities of these conditions. Furthermore, the COVID-19 pandemic and influenced dental intervention [14,15] and could significantly impact vulnerable children [16], disrupting school-based oral interventions and imposing global economic burdens that affect treatment decisions, thereby exacerbating disparities, particularly among disadvantaged locations [17,18]. While existing literature has focused on global trends in oral disease burden among 10–24-year-olds as a single group [13], there is a paucity of research examining lifecycle-specific disparities in children, particularly in differential COVID-19 impacts on oral epidemiological status and socioeconomic inequalities [7], leaving a gap in lifecycle-specific evidence to address post-pandemic disparities [17,18].

Therefore, this study utilizes GBD 2021 data to complement existing research by offering a detailed analysis of the burden, trends, and inequalities of oral disorders in children, adolescents, and young adults from 1990 to 2021. It primarily focuses

on years lived with disability (YLDs), followed by incidence and prevalence metrics. The young populations (0–39 years) were further categorized into three subgroups: children (0–9 years), adolescents (10–19 years), and young adults (20–39 years). This stratification reflects lifecycle-specific oral health transitions and addresses specific challenges within each subgroup [2]:

1) Children: Predominance of deciduous caries; emergence of permanent caries [4].

2) Adolescents: Escalation of permanent caries; early onset of periodontal disease [10].

3) Young adults: Advanced caries progression, periodontal damage, and early edentulism [2].

By bridging pediatric to adult oral health frameworks, this study aims to provide comprehensive global disease burden estimates for oral health. Enhancing our understanding of the critical health challenges faced by children, adolescents, and young adults will inform the development of targeted policies and practices for both the public and practitioners.

## 2. Materials and methods

Using Joinpoint regression, decomposition analysis, and inequality analyzes (the slope index and concentration index), this study systematically assesses the burden of oral disorders across three subgroups: children (0–9 years), adolescents (10–19 years), and young adults (20–39 years), with a primary focus on YLDs, followed by incidence and prevalence.
Specifically, these methodologies are used to:

1. Joinpoint regression: Identify the burden of oral diseases and the temporal trends, including significant change points.

2. Decomposition analysis: Decompose the drivers of burden changes into demographic and epidemiological components,

3. Inequality analyses: Quantify absolute and relative socioeconomic inequalities to prioritize equitable resource allocation.

All outcomes are presented at the general (0–39 years) and subgroup levels. Fig 1 is a flowchart of this study.

### 2.1. Data source

To determine the global burden of oral disorders among young populations, this study utilizes the 2021 GBD data on 5 specific causes: untreated caries in deciduous teeth, untreated caries in permanent teeth, periodontal diseases, edentulism, and other oral disorders [1]. Case definitions are consistent with previous references in S1 Text [2,3]. The full-time series from 1990 to 2021 was recalculated to produce comparable estimates. Detailed GBD methodology is available in earlier references, and strategies for oral disorders are provided in S1 Text [2,4,19]. This analysis adheres to the Guidelines for Accurate and Transparent Health Estimates Reporting (GATHER) (S2 Text). GBD 2021 data were extracted from the Global Health Data Exchange (https://ghdx.healthdata.org). Data on countries and territories categorized by sociodemographic index (SDI) quintiles were also obtained (Table A in S3 Text) [20]. The SDI is a composite indicator strongly correlated with a location's sociodemographic development status and health outcomes, calculated as the geometric mean of three indices: total fertility rate among those under 25, mean education level for individuals aged 15 and older, and lag-distributed income per capita. The SDI ranges from 0 (lowest) to 1 (highest) (S1 Text).
Based on the GBD world standard population (Fig A and Table B in S3 Text), we utilized GBD 2021 data on oral disorders to calculate age-standardized rates categorized by cause, year (1990–2021), sex (female, male, and both), SDI quintile (low-, low-middle-, middle-, high-middle-, and high-SDI quintiles), and country or territory (Table C-F in S3 Text) at both general and subgroup levels. The age-standardized rates were calculated using the following formula [21]:

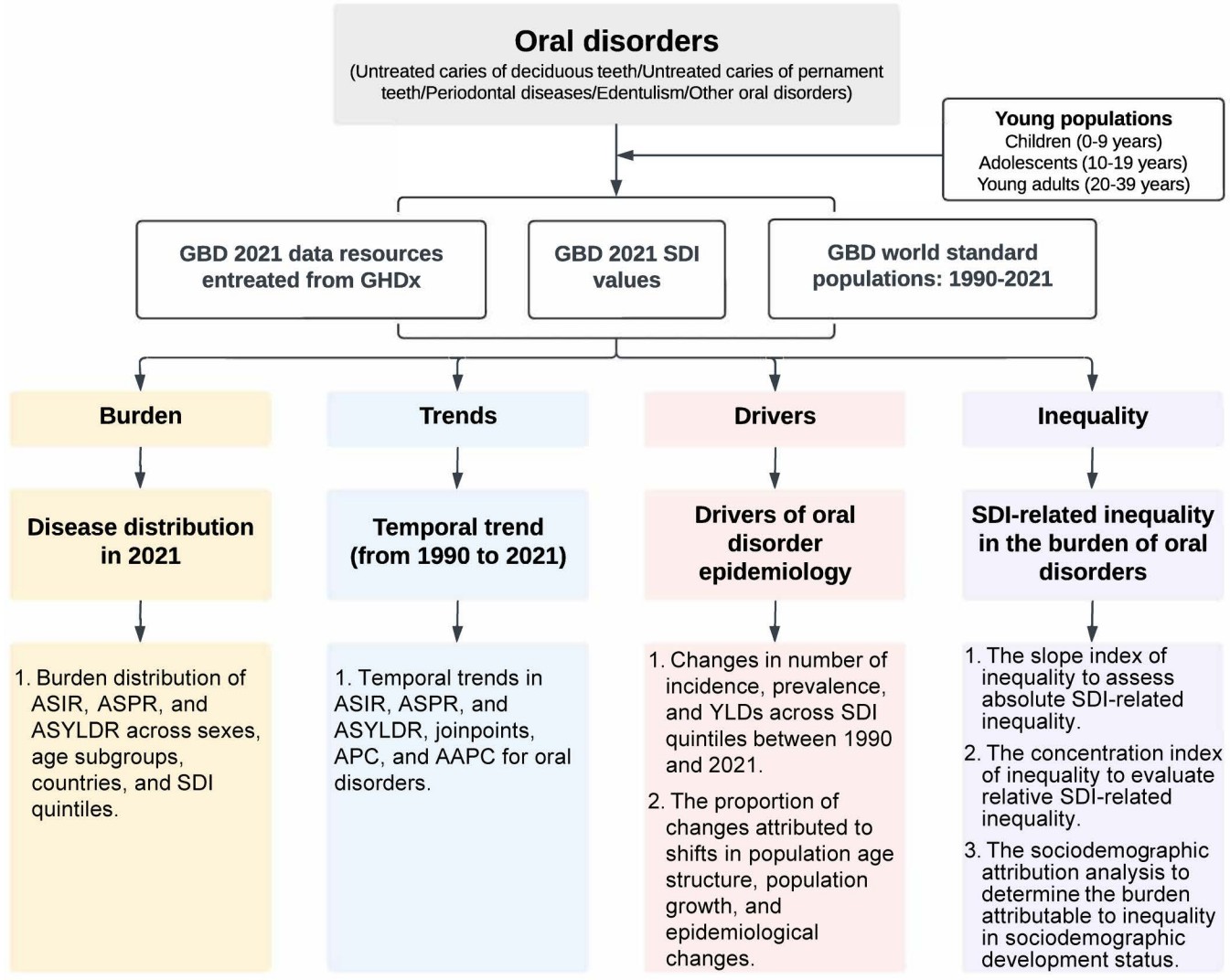

**Fig 1. Flowchart of this study.** GBD, Global Burden of Diseases; GHDx, the Global Health Data Exchange; SDI, sociodemographic index; ASIR, age-standardized incidence rate; ASPR, age-standardized prevalence rate; ASYLDR, age-standardized years lived with disability rate; YLD, year lived with disability; APC, annual percent change; AAPC, average annual percent change.

$$ASR = \left( \frac{\sum_{i=1}^{n} (w_i \cdot r_i)}{\sum_{i=1}^{n} w_i} \right) \times C$$

ASR, age-standardized rates; $w_i$, weight for age group $i$, derived from the GBD world standard population; $r_i$, the crude rate for age group $i$; $C$, the constant to scale the rate (100000 in this study).

Subgroup-level age standardization accounts for variations in population size and age structure, enabling valid comparisons across regions, periods, and subgroups with differing demographic compositions. Dismod-MR 2.1 is the tool used for estimating the incidence and prevalence of non-fatal diseases [19,22]. Given that oral disorders rarely result in death, the GBD 2021 study assumed years of life lost (YLLs) for these disorders to be zero. Consequently, the estimates

of disability-adjusted life years, which comprise the sum of YLLs and YLDs, equate to YLDs. YLDs were calculated by multiplying the prevalence, disability duration, proportion with symptoms, and disability weight for each age, country, sex, and year. More details on the GBD estimation framework are found in the S1 Text. The study also includes the number of health indicators across different years, sexes, and SDI quintiles.

## 2.2. Joinpoint regression

A Joinpoint regression model (version 4.9.1.0; Joinpoint Regression Program, National Cancer Institute) was used to assess temporal trends in the burden of oral disorders from 1990 to 2021, estimating the average annual percentage change (AAPC) with a 95% confidence interval (CI) for health indicators (S1 Text) [23]. The AAPC summarizes the trend over a prespecified interval, calculated as a weighted average of annual percentage change (APC), with weights equal to the length of the APC interval. The model identifies the best fit for joinpoints, indicating statistically significant trend changes. A maximum of five joinpoints was allowed for the algorithmic recommendations from the National Cancer Institute (https://surveillance.cancer.gov/help/joinpoint/setting-parameters/method-and-parameters-tab/number-of-joinpoints), with model selection conducted using the Monte Carlo Permutation method. Trends were classified as increasing (worsening), decreasing (improving), or stable/level based on whether the APC significantly deviated from 0.

## 2.3. Decomposition analysis

Decomposition analysis quantitatively analyzes the drivers of change in health indicators, which was performed using the R programming environment (version 4.2.0; R Core Team, Vienna, Austria). To analyze the epidemiology of oral disorders, changes in burden among young populations from 1990 to 2021 were decomposed into 3 explanatory components: changes attributed to shifts in population age structure, population growth, and epidemiological changes (referred to here as age-standardized rates) [24]. The effect of each component was assessed counterfactually by varying one component per period while keeping the other two constant (S1 Text) [25].

## 2.4. Sociodemographic inequality analysis

The distributive inequality of oral disorder burdens across different SDI quintiles was measured using three metrics: the slope index of inequality to assess absolute inequality, the concentration index of inequality to evaluate relative inequality, and sociodemographic attribution analysis to determine the burden attributable to inequality in sociodemographic development [8,22,26]. The slope index of inequality was computed by regressing national-level age-standardized rates of oral disorders on an SDI-based relative rank, defined as the midpoint of the cumulative population fraction ranked by SDI quintiles. The value could be understood as an estimate of the difference in a given health indicator between the locations with the highest and lowest SDIs. The concentration index of inequality was calculated by fitting a Lorenz concentration curve to the observed cumulative fraction of the population ranked by SDI and oral disorder burdens, followed by numerical integration of the area under the curve. A negative value of the concentration index indicates that the health indicator is concentrated among poorer locations, while a positive value suggests concentration among wealthier locations. When there is no inequality, the concentration index equals zero. Its theoretical maximum is ± 1, with values ranging from 0.2 to 0.3 considered to represent a reasonably high level of relative inequality [27]. The inequality analyses were performed using the HEAT plus software (version 6.0; World Health Organization, Geneva) [28], in accordance with the WHO Handbook on Health Inequality Monitoring [27]. Please see S1 Text for detailed methodology.

## 2.5. Data availability

Data sources and codes used in the Global Burden of Disease Study 2021 are publicly available at http://ghdx.healthdata.org/gbd-results-tool.

## 3. Results

### 3.1. YLDs associated with oral disorders among young populations in 2021

The global distribution of oral disorder burdens in 2021 is illustrated in Fig 2 and Fig B-D in S3 Text. Globally, periodontal disease contributed most to oral disorder-related YLDs among young populations in 2021, with an age-standardized YLD rate (ASYLDR) of 58.89 (22.95-131.1) (per 100,000 populations) (Table 1). Subgroup analyses revealed distinct epidemiological patterns: untreated caries in deciduous teeth dominated the burden in children (0–9 years), with an ASYLDR of 14.0 (6.2–27.29) and untreated caries in permanent teeth prevailed among adolescents (10–19 years) at 27.27 (10.41–56.74) (per 100,000 populations). Young adults (20–39 years) experienced a high burden of ASYLDR from untreated

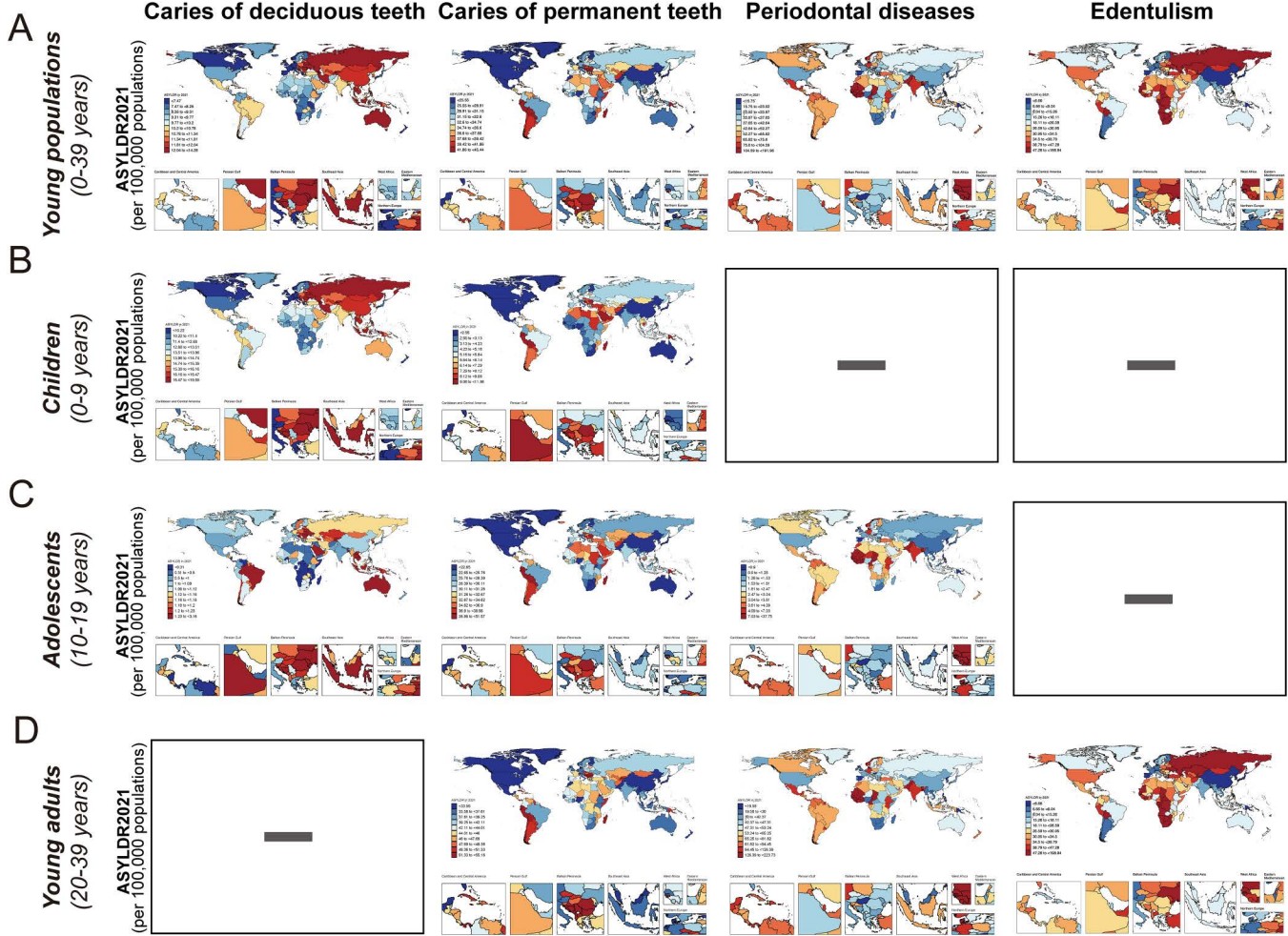

**Fig 2. The distribution of age-standardized YLD rates per 100,000 populations of oral disorders for 204 countries and territories in 2021. (A)** The distribution of age-standardized YLD rates per 100,000 populations in young population (0-39 years) for oral disorders across 204 countries and territories in 2021. **(B)** The distribution of age-standardized YLD rates per 100,000 populations in children (0-9 years) for oral disorders across 204 countries and territories in 2021. **(C)** The distribution of age-standardized YLD rates per 100,000 populations in adolescents (10-19 years) for oral disorders across 204 countries and territories in 2021. **(D)** The distribution of age-standardized YLD rates per 100,000 populations in young adults (20-39 years) for oral disorders across 204 countries and territories in 2021. ASYLDR, age-standardized years lived with disability rate. Republished from https://www.resdc.cn/data.aspx?DATAID=205 under a CC BY license, with permission from the Data Center for Resources and Environmental Sciences (RESDC), original copyright 2025.

**Table 1. The age-standardized incidence, prevalence, and YLD rates per 100,000 populations of oral disorders, across different sexes, age groups, and SDI quintiles, in 1990 and 2021, as well as the AAPCs in the period 1990–2021.**

| Characteristic | 1990 | | | 2021 | | | AAPC | | |
|---|---|---|---|---|---|---|---|---|---|
| | ASIR (per 100,000), Rate (95% UI) | ASPR (per 100,000), Rate (95% UI) | ASYLDR (per 100,000), Rate (95% UI) | ASIR (per 100,000), Rate (95% UI) | ASPR (per 100,000), Rate (95% UI) | ASYLDR (per 100,000), Rate (95% UI) | ASIR, No. (95% CI) | ASPR, No. (95% CI) | ASYLDR, No. (95% CI) |
| **Caries of deciduous teeth** | | | | | | | | | |
| Global | 67151.61 (40746.58-103682.3) | 28174.18 (20233.66-36437.02) | 10.74 (4.39-21.27) | 62261.07 (41905.15-89708.76) | 26566.05 (20396.01-33066.03) | 10.16 (4.42-20.02) | -0.23 (-0.28 to -0.18) | -0.17 (-0.21 to -0.13) | -0.16 (-0.2 to -0.12) |
| **Sex** | | | | | | | | | |
| Female | 66755.27 (40706.58-102641.89) | 27994.75 (20237.75-36080.43) | 10.68 (4.39-21.04) | 61743.74 (41709.07-88345.38) | 26410.87 (20324.99-32874.05) | 10.1 (4.39-19.92) | -0.25 (-0.27 to -0.24) | -0.18 (-0.21 to -0.14) | -0.17 (-0.2 to -0.14) |
| Male | 67528.07 (40810.33-104898.1) | 28343.26 (20219.31-36751.31) | 10.8 (4.39-21.55) | 62746.73 (42172.97-90639.04) | 26711.32 (20443.09-33288.97) | 10.22 (4.45-20.11) | -0.2 (-0.25 to -0.15) | -0.17 (-0.21 to -0.14) | -0.16 (-0.2 to -0.13) |
| **Age subgroup** | | | | | | | | | |
| Children | 81334.81 (55659.16-105091.6) | 38260.56 (28553.9-48196.04) | 14.59 (6.13-28.47) | 79162.97 (58508.57-98800.15) | 36597.61 (28857.24-44513.54) | 14 (6.2-27.29) | -0.07 (-0.12 to -0.03) | -0.13 (-0.17 to -0.09) | -0.12 (-0.15 to -0.09) |
| Adolescents | 122094.32 18777.44 (4205.37-52347.88) | 3164.16 (1045.25-5557.4) | 1.21 (0.3-2.87) | 116346.23 13133.56 (2880.95-36319.75) | 2389.82 (969.01-4158.29) | 0.91 (0.27-2.13) | -1.11 (-1.27 to -0.94) | -0.86 (-0.93 to -0.8) | -0.86 (-0.93 to -0.79) |
| **Sociodemographic index** | | | | | | | | | |
| Low SDI | 67446.95 (40643.52-103354.72) | 27089.6 (19308.69-35211.62) | 10.28 (4.23-20.2) | 58395.89 (39749.16-83223.58) | 24661.72 (18732.19-30991.19) | 9.41 (4.1-18.73) | -0.46 (-0.52 to -0.4) | -0.3 (-0.35 to -0.24) | -0.28 (-0.32 to -0.23) |
| Low-middle SDI | 68886.24 (40366.07-108346.93) | 27893.38 (19431.3-36578.7) | 10.61 (4.31-21.06) | 60391.4 (41125.01-85720.76) | 25687.6 (19450.81-32236.46) | 9.82 (4.26-19.75) | -0.41 (-0.45 to -0.36) | -0.25 (-0.29 to -0.22) | -0.24 (-0.28 to -0.2) |
| Middle SDI | 67724.88 (40909.05-104345.18) | 29537.1 (21472.71-38022.31) | 11.28 (4.61-22.43) | 65823.34 (43727.82-96019.36) | 29422.9 (22821.37-36493.35) | 11.27 (4.83-22.27) | -0.09 (-0.12 to -0.06) | 0 (-0.03 to 0.03) | 0.01 (-0.02 to 0.04) |
| High-middle SDI | 69322.78 (42765.22-106148.83) | 30274.75 (21981.55-38705.29) | 11.59 (4.8-23.07) | 68399.3 (42326.88-104303.25) | 29252.41 (21741.01-37275.82) | 11.22 (4.65-22.28) | -0.05 (-0.08 to -0.02) | -0.11 (-0.17 to -0.05) | -0.1 (-0.16 to -0.04) |
| High SDI | 57284.16 (35375.66-87716.59) | 22912.67 (16911.31-29427.72) | 8.77 (3.66-17.41) | 58202.28 (38570.71-84969.78) | 22040.88 (16369.35-28331.23) | 8.46 (3.51-16.35) | 0.06 (-0.01 to 0.13) | -0.1 (-0.23 to 0.04) | -0.09 (-0.22 to 0.04) |
| **Caries of permanent teeth** | | | | | | | | | |
| Global | 36279.45 (26636.73-46709.17) | 29782.7 (20127.5-41986.59) | 29.64 (11.61-59.77) | 39079.88 (29317.45-49476.57) | 28989.1 (20361.45-39942.86) | 28.86 (11.38-57.95) | 0.24 (0.22 to 0.26) | -0.09 (-0.11 to -0.07) | -0.09 (-0.11 to -0.06) |
| **Sex** | | | | | | | | | |
| Female | 36693.04 (27144.28-46977.61) | 30195.33 (20529.23-42341.27) | 29.95 (11.79-60.26) | 39175.06 (29405.19-49687.56) | 29287.66 (20536.18-40356.46) | 29.04 (11.48-58.23) | 0.21 (0.19 to 0.23) | -0.1 (-0.13 to -0.08) | -0.1 (-0.13 to -0.08) |
| Male | 35879.62 (26138.07-46445.18) | 29383.3 (19726.57-41616.1) | 29.34 (11.43-59.31) | 38989.49 (29245.29-49284.51) | 28703.45 (20171.14-39607.33) | 28.7 (11.28-57.68) | 0.27 (0.26 to 0.29) | -0.08 (-0.1 to -0.05) | -0.07 (-0.1 to -0.04) |

*(Continued)*

| Characteristic | 1990 | | | 2021 | | | AAPC | | |
|---|---|---|---|---|---|---|---|---|---|
| | ASIR (per 100,000), Rate (95% UI) | ASPR (per 100,000), Rate (95% UI) | ASYLDR (per 100,000), Rate (95% UI) | ASIR (per 100,000), Rate (95% UI) | ASPR (per 100,000), Rate (95% UI) | ASYLDR (per 100,000), Rate (95% UI) | ASIR, No. (95% CI) | ASPR, No. (95% CI) | ASYLDR, No. (95% CI) |
| **Age subgroup** | | | | | | | | | |
| **Children** | 8810.66 (4369.22-15228.83) | 4655.89 (2354.31-7845.88) | 4.69 (1.54-11.03) | 7949.2 (3971.38-14080.86) | 3951.78 (1977.85-6581.34) | 3.99 (1.3-9.44) | -0.3 (-0.34 to -0.26) | -0.49 (-0.56 to -0.41) | -0.48 (-0.56 to -0.4) |
| **Adolescents** | 36648.08 (26798.16-47359.65) | 10395.21 (6501.11-14942.37) | 28.48 (10.92-60.26) | 40511.71 (29917.58-52157.26) | 27216.68 (18362.55-37915.75) | 27.27 (10.41-56.74) | 0.33 (0.28 to 0.38) | -0.14 (-0.18 to -0.11) | -0.14 (-0.18 to -0.11) |
| **Young adults** | 42049.27 (32337.01-51443.98) | 37200.01 (25976.02-51437.21) | 36.87 (14.79-71.66) | 45717.71 (35902.78-54693.92) | 36868.14 (26868.65-49798.48) | 36.57 (14.79-71.3) | 0.28 (0.25 to 0.31) | -0.03 (-0.05 to -0.01) | -0.02 (-0.04 to 0) |
| **Sociodemographic index** | | | | | | | | | |
| **Low SDI** | 37929.75 (28957.31-47788.63) | 33348.78 (24091.48-44284.66) | 32.97 (13.36-65.7) | 38399.8 (29275.48-48414) | 31978.8 (22992.58-42807.2) | 31.74 (12.88-63.24) | 0.03 (0.02 to 0.05) | -0.14 (-0.17 to -0.12) | -0.13 (-0.16 to -0.11) |
| **Low-middle SDI** | 39668.71 (28926.75-51062.38) | 31410.1 (21334.31-44040.56) | 31.14 (12.14-62.87) | 40086.55 (29697.07-51047.91) | 29642.63 (20375.11-41497.03) | 29.47 (11.43-59.45) | 0.03 (0.01 to 0.06) | -0.19 (-0.23 to -0.14) | -0.18 (-0.22 to -0.14) |
| **Middle SDI** | 34231.9 (24339.77-44977.92) | 29255.74 (19163.31-42162.73) | 29.16 (11.26-59.68) | 39231.58 (29071.52-49933.21) | 28617.76 (19748.8-39931.58) | 28.54 (11.2-57.88) | 0.47 (0.4 to 0.54) | -0.07 (-0.1 to -0.03) | -0.06 (-0.1 to -0.02) |
| **High-middle SDI** | 32973.37 (23529.42-43224.61) | 29839.14 (19268.6-42977.92) | 29.79 (11.45-60.8) | 37256.14 (27200.41-48101.37) | 28300.95 (19507.51-39447.72) | 28.26 (11.05-57.24) | 0.39 (0.34 to 0.43) | -0.17 (-0.23 to -0.1) | -0.16 (-0.23 to -0.1) |
| **High SDI** | 40117.29 (31294.19-49376.95) | 26946.54 (19297.5-36935.62) | 26.84 (10.72-53.76) | 39968.46 (31279.53-49242.42) | 25307.91 (18681.66-33819.14) | 25.19 (10.28-50.61) | -0.01 (-0.03 to 0.01) | -0.2 (-0.23 to -0.188) | -0.2 (-0.23 to -0.17) |
| **Periodontal diseases** | | | | | | | | | |
| **Global** | 1122.83 (587.4-1742.04) | 8364.63 (5197.6-12112.46) | 55.21 (20.29-119.47) | 1185.12 (744.5-1720.65) | 9063.06 (6368.42-12231.02) | 59.89 (22.95-131.1) | 0.18 (0.16 to 0.21) | 0.26 (0.111 to 0.35) | 0.26 (0.18 to 0.35) |
| **Sex** | | | | | | | | | |
| **Female** | 1091.73 (577.26-1686.78) | 8125.29 (5061.44-11749.18) | 53.33 (19.66-115.28) | 1170.99 (735.59-1700.9) | 8941.8 (6281.44-12097.97) | 58.74 (22.56-128.32) | 0.24 (0.21 to 0.27) | 0.31 (0.26 to 0.36) | 0.31 (0.27 to 0.36) |
| **Male** | 1153.17 (595.23-1794.24) | 8597.84 (5322.25-12471.71) | 57.04 (20.91-123.58) | 1198.86 (754.47-1740.66) | 9181.54 (6450.99-12365.04) | 61.01 (23.35-133.83) | 0.13 (0.09 to 0.18) | 0.21 (0.15 to 0.27) | 0.21 (0.16 to 0.27) |
| **Age subgroup** | | | | | | | | | |
| **Adolescents** | 186.74 (91.3-334.54) | 540.84 (264.69-972.03) | 3.62 (1.02-9.1) | 209.19 (125.13-328.35) | 626.09 (387.37-943.72) | 4.17 (1.42-9.05) | 0.35 (0.17 to 0.52) | 0.46 (0.33 to 0.59) | 0.45 (0.31 to 0.59) |
| **Young adults** | 1328.63 (698.8-2035.13) | 10395.21 (6501.11-14942.37) | 68.58 (25.39-147.67) | 1395.26 (880.2-2011.34) | 11239.74 (7928.68-15110.76) | 74.25 (28.57-162.59) | 0.17 (0.15 to 0.2) | 0.25 (0.18 to 0.32) | 0.25 (0.19 to 0.32) |
| **Sociodemographic index** | | | | | | | | | |
| **Low SDI** | 1734.91 (946.88-2549.58) | 15323.73 (10032.1-21050.95) | 100.76 (37.19-215.53) | 1270.69 (814.41-1805.89) | 11025.85 (8066.72-14344.74) | 72.71 (28.68-155.09) | -1 (-1.09 to -0.91) | -1.07 (-1.17 to -0.97) | -1.06 (-1.15 to -0.96) |

*(Continued)*

| Characteristic | 1990 | | | 2021 | | | AAPC | | |
|---|---|---|---|---|---|---|---|---|---|
| | ASIR (per 100,000), Rate (95% UI) | ASPR (per 100,000), Rate (95% UI) | ASYLDR (per 100,000), Rate (95% UI) | ASIR (per 100,000), Rate (95% UI) | ASPR (per 100,000), Rate (95% UI) | ASYLDR (per 100,000), Rate (95% UI) | ASIR, No. (95% CI) | ASPR, No. (95% CI) | ASYLDR, No. (95% CI) |
| **Low-middle SDI** | 1468.03 (791.2-2206.42) | 12138.74 (7774.18-17075.71) | 79.95 (29.39-171.63) | 1488.37 (944.04-2104.3) | 12497.28 (8910.99-16380.95) | 82.54 (31.38-181) | 0.04 (-0.01 to 0.08) | 0.09 (0.02 to 0.17) | 0.11 (0.03 to 0.19) |
| **Middle SDI** | 1057.73 (545.49-1659.77) | 7520.65 (4553.68-11111.52) | 49.7 (18.02-106.98) | 1125.54 (712.69-1621.53) | 8267.89 (5825.4-11042.38) | 54.69 (21.24-119.73) | 0.22 (0.18 to 0.26) | 0.32 (0.24 to 0.39) | 0.32 (0.24 to 0.4) |
| **High-middle SDI** | 873.75 (427.63-1436.16) | 5678.96 (3269.99-8786.85) | 37.56 (13.3-84.26) | 897.87 (527.75-1447.66) | 5971.04 (3876.57-9118.35) | 39.5 (14.49-90.77) | 0.11 (0.04 to 0.17) | 0.18 (0.12 to 0.24) | 0.18 (0.12 to 0.24) |
| **High SDI** | 826.22 (429.87-1316.26) | 5912.38 (3760.7-8636.05) | 39.11 (14.35-84.61) | 831.52 (474.61-1343.53) | 5787.48 (3656.09-8799.97) | 38.23 (13.93-88.12) | 0.02 (-0.07 to 0.11) | -0.06 (-0.17 to 0.05) | -0.06 (-0.188 to 0.05) |
| **Edentulism** | | | | | | | | | |
| **Global** | 85.66 (46.68-139.41) | 709.56 (433.15-1018.48) | 20.38 (10.51-33.58) | 89.21 (55.6-133.08) | 752.26 (522.08-1027.41) | 21.6 (12.17-33.84) | 0.16 (-0.05 to 0.38) | 0.22 (-0.02 to 0.45) | 0.21 (-0.02 to 0.45) |
| **Sex** | | | | | | | | | |
| **Female** | 95.13 (52.06-154.33) | 788.02 (482.81-1124.68) | 22.49 (11.71-36.91) | 97.9 (61.35-145.06) | 827.03 (579.4-1124.99) | 23.58 (13.36-36.66) | 0.12 (-0.04 to 0.29) | 0.19 (-0.09 to 0.47) | 0.18 (-0.13 to 0.5) |
| **Male** | 76.4 (41.39-124.52) | 633.03 (383.91-914.84) | 18.31 (9.28-30.3) | 80.71 (49.66-121.33) | 679.06 (462.67-933.93) | 19.67 (10.96-31.19) | 0.18 (-0.21 to 0.57) | 0.25 (-0.11 to 0.61) | 0.25 (-0.1 to 0.61) |
| **Age subgroup** | | | | | | | | | |
| **Young adults** | 85.66 (46.68-139.41) | 709.56 (433.15-1018.48) | 20.38 (10.51-33.58) | 89.21 (55.6-133.08) | 752.26 (522.08-1027.41) | 21.6 (12.17-33.84) | 0.16 (-0.05 to 0.38) | 0.22 (-0.02 to 0.45) | 0.21 (-0.02 to 0.45) |
| **Sociodemographic index** | | | | | | | | | |
| **Low SDI** | 108.67 (59.52-172.08) | 1155.37 (723.22-1631.6) | 32.96 (17.27-53.98) | 103.43 (59.83-159.97) | 1090.35 (710.91-1512.41) | 31.25 (17.14-50.37) | -0.24 (-0.86 to 0.38) | -0.27 (-0.96 to 0.42) | -0.25 (-0.94 to 0.44) |
| **Low-middle SDI** | 89.93 (47.72-147.53) | 832.25 (497.8-1204.45) | 23.84 (12.04-39.68) | 77.39 (47.55-116.5) | 709.41 (485.44-971.87) | 20.37 (11.38-32.36) | -0.48 (-1.26 to 0.31) | -0.51 (-1.37 to 0.35) | -0.51 (-1.37 to 0.36) |
| **Middle SDI** | 82.2 (44.68-133.67) | 685.07 (420.68-983.8) | 19.71 (10.09-32.39) | 88.98 (56.04-131.62) | 741.41 (524.56-998.97) | 21.29 (12.16-32.93) | 0.31 (0.09 to 0.52) | 0.3 (0.06 to 0.555) | 0.3 (0.04 to 0.56) |
| **High-middle SDI** | 80.88 (43.69-130.6) | 646.48 (394.84-931.01) | 18.62 (9.34-30.8) | 78.02 (49.56-115.12) | 607.5 (433.44-824.14) | 17.5 (10.13-27.04) | -0.11 (-0.15 to -0.06) | -0.2 (-0.45 to 0.05) | -0.2 (-0.46 to 0.05) |
| **High SDI** | 78.57 (42.77-129.08) | 495.79 (299.72-735.28) | 14.26 (7.11-23.81) | 103.36 (59.44-162.94) | 686.84 (440.25-992.86) | 19.7 (10.31-32.23) | 0.9 (0.72 to 1.08) | 1.12 (0.82 to 1.43) | 1.12 (0.81 to 1.42) |

The age-standardized incidence, prevalence and YLD rates (values in brackets correspond to the 95% uncertainty interval) per 100,000 populations, along with the average annual percent changes (values in brackets correspond to the 95% confidence interval), of untreated caries of deciduous teeth, untreated caries of permanent teeth, periodontal diseases, edentulism, and other oral disorders, across different sexes, age subgroups, and SDI quintiles, in 1990 and 2021.

YLD, year lived with disability; AAPC, average annual percent change; SDI, sociodemographic index; ASIR, age-standardized incidence rate; ASPR, age-standardized prevalence rate; ASYLDR, age-standardized years lived with disability rate; UI, uncertainty interval; CI, confidence interval.

caries in permanent teeth [36.57 (14.79–71.3)], periodontal diseases [74.25 (28.57–162.59)] and edentulism [21.6 (12.17-33.84)]. Periodontal diseases and edentulism were absent in children, while edentulism was not observed in adolescents, and caries of deciduous teeth were unreported in young adults. (Table 1).

### 3.2. Temporal trends in oral disorder burden of YLDs from 1990 to 2021

From 1990 to 2021, the global YLD number for oral disorders increased from 4.45 (2.40–7.46) to 6.22 (3.40–10.57) million among young populations (Fig E-G and Table G-H in S3 Text). Temporal trends in YLD changes varied by subgroup: the ASYLDR of untreated deciduous caries in children declined from 1990 to 2009, stabilized until 2016, and then rebounded slightly (2016–2021), while adolescents experienced a continuous 30-year decline. Untreated permanent caries ASYLDRs across all subgroups exhibited a similar trajectory—initial declines (1990–2000), a peak in 2005, and subsequent reductions, although rates remained below 1990 levels despite a slight uptick. Trends in periodontal disease diverged between adolescents and young adults. The ASYLDR of periodontal disease among adolescents and young adults in 2021 exceeded 1990 levels. Firstly, both groups exhibited similar trends: rates rose initially, declined until approximately 2010, and then increased again, peaking around 2014. Divergence emerged after 2015, with adolescents experiencing a decline in ASYLDR from 2015 to 2019, followed by a slight rebound (2019–2021), while young adults saw a persistent increase through 2021. For edentulism in young adults, ASYLDR declined to its lowest point in 2000, peaked in 2015, and subsequently decreased through 2021, yet remained elevated compared to 1990 levels (Fig 3). Comprehensive ASYLDR data across countries and territories, along with AAPCs, are provided in Table I-R in S3 Text.

### 3.3. Drivers of oral disorder epidemiology

Decomposition analyzes identified distinct demographic and epidemiological drivers of changes in oral disorder burdens between 1990 and 2021 (Fig 4). Globally, population growth was the primary contributor to YLD increases, accounting for 219.8% of the rise in untreated deciduous caries, 102.8% for untreated permanent caries, 64.7% for periodontal diseases, and 72.4% for edentulism. Absolute YLD increases were pronounced for untreated deciduous teeth caries in children (+13626.59 YLDs), untreated permanent caries in adolescents (+51115.62 YLDs), permanent caries (+239617.66 YLDs), and periodontal diseases (+670,667.78 YLDs) in young adults. Subgroup and SDI-stratified analyses further clarified in lower-SDI locations, population growth was the primary driver of YLD increases across all disorders. Conversely, population growth contributed to reductions in YLDs for untreated deciduous and permanent caries in children and adolescents in high-SDI quintiles, for example, an 85.49% reduction of YLDs of deciduous caries in children. Epidemiological change (reflected in ASYLDR shifts) emerged as the second driver globally. For caries, it reduced YLDs across most subgroups and SDI quintiles, with pronounced effects observed in children's permanent caries burden in high-SDI locations (-7130.18 YLDs) and adolescents' deciduous caries burden in low-SDI locations (-2625.8 YLDs). Population age structure exerted a notable influence (>10%) only on periodontal disease and edentulism in young adults, driving increased YLD burdens. Detailed numerical and proportional contributions of the drivers across all subgroups and SDI quintiles are provided in Table G-H in S3 Text.

### 3.4. SDI-related inequality in oral disorder YLD burdens

Analyses of absolute (slope index) and relative (concentration index) inequalities revealed different socioeconomic patterns across oral disorders. For untreated deciduous and permanent caries and edentulism, no significant absolute or relative SDI-related inequalities were observed in the ASYLDR generally or within subgroups, with no significant difference between the results of 1990 and 2021. Periodontal disease burdens exhibited pronounced socioeconomic disparities. The slope index indicated heavier ASYLDR burdens in lower-SDI locations compared to higher-SDI locations in both 1990 [slope index: -61.39 (-76.42 to -46.37)] and 2021 [slope index: -16.71 (-30.05 to -3.37)], with the absolute inequality gap narrowed over time (Fig 5). Subgroup analyses supported the general findings of SDI-related inequalities

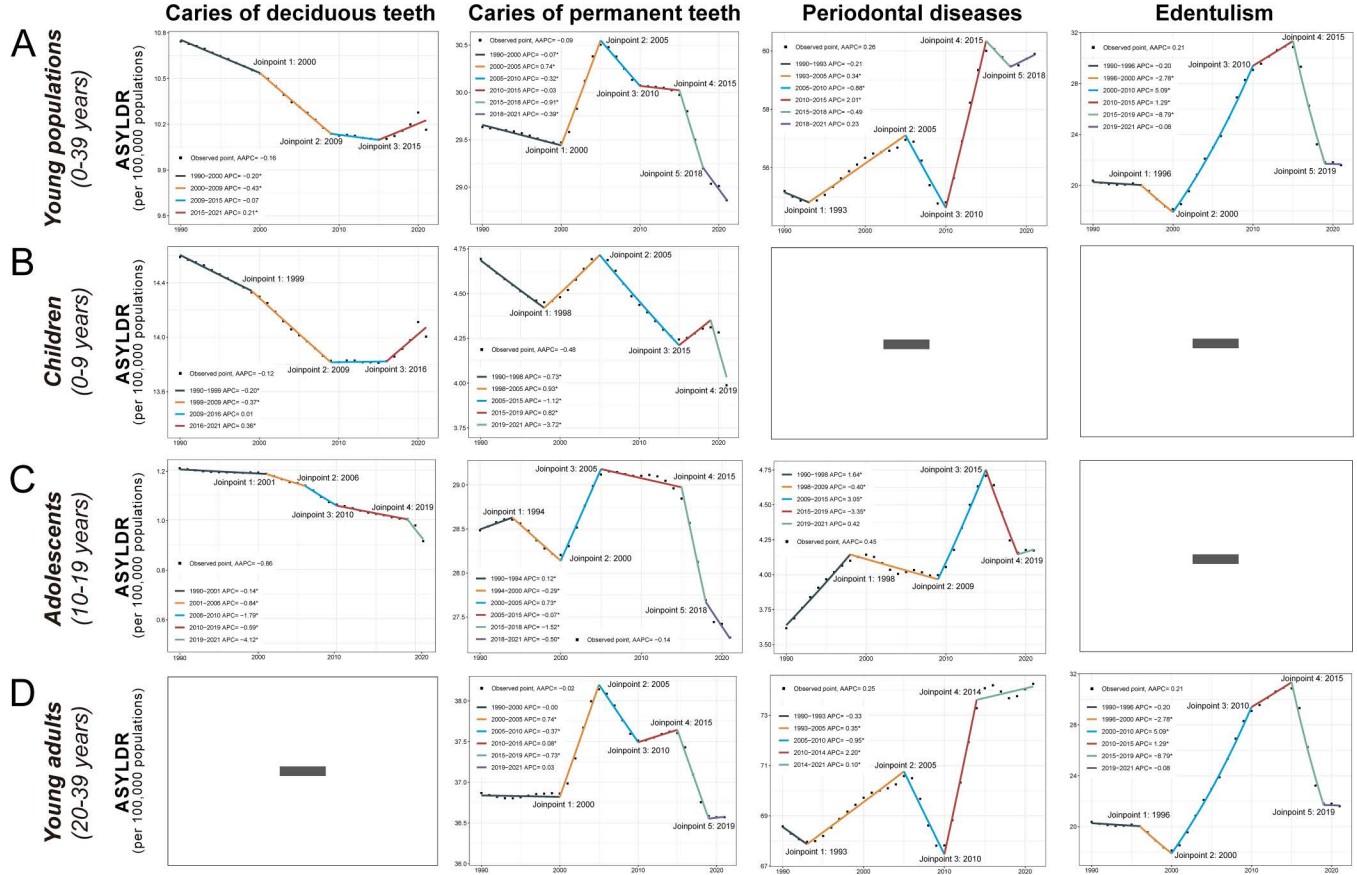

**Fig 3. The temporal trends in age-standardized YLD rates per 100,000 populations of oral disorders from 1990 to 2021, and joinpoints, annual percentage changes, and average annual percentage changes in the period 1990-2021. (A)** The temporal trends in age-standardized YLD rates per 100,000 populations in young population (0-39 years) for oral disorders from 1990 to 2021. **(B)** The temporal trends in age-standardized YLD rates per 100,000 populations in children (0-9 years) for oral disorders from 1990 to 2021. **(C)** The temporal trends in age-standardized YLD rates per 100,000 populations in adolescents (10-19 years) for oral disorders from 1990 to 2021. **(D)** The temporal trends in age-standardized YLD rates per 100,000 populations in young adults (20-39 years) for oral disorders from 1990 to 2021. ASYLDR, age-standardized years lived with disability rate; APC, annual percentage changes; AAPC, average annual percentage changes.

and further stratified the patterns. The absolute inequality gap in the ASYLDRs of periodontal disease was wider among young adults [slope index: -74.82 (-93.65 to -56.00) in 1990; -20.04 (-36.66 to -3.43) in 2021] than in adolescents [slope index: -6.11 (-7.20 to -5.02) in 1990; -1.86 (-2.64 to -1.08) in 2021]. The concentration index corroborated the results of the slope index, indicating a disproportionate concentration of ASYLDR for periodontal disease in lower-SDI locations in both 1990 [concentration index: -0.2 (-0.23 to -0.16)] and 2021[concentration index: -0.17 (-0.21 to -0.13)] (Fig 6). Notably, adolescents exhibited greater relative inequalities than young adults in ASYLDR of periodontal disease, with a concentration index of -0.35 (-0.42 to -0.28) in 1990 and -0.29 (-0.36 to -0.21) in 2021 for adolescents, compared to -0.19 (-0.23 to -0.15) and -0.17 (-0.21 to -0.13) for young adults. Despite persistent disparities, improvements in the relative inequality of periodontal disease were observed from 1990 to 2021.

Although secondary to YLDs in policy relevance, the incidence and prevalence metrics (Fig H-O in S3 Text), outcomes of other oral disorders (Fig P-Q in S3 Text), and results of the sociodemographic attribution analysis are presented in S4 Text to ensure methodological transparency and support reproducibility.

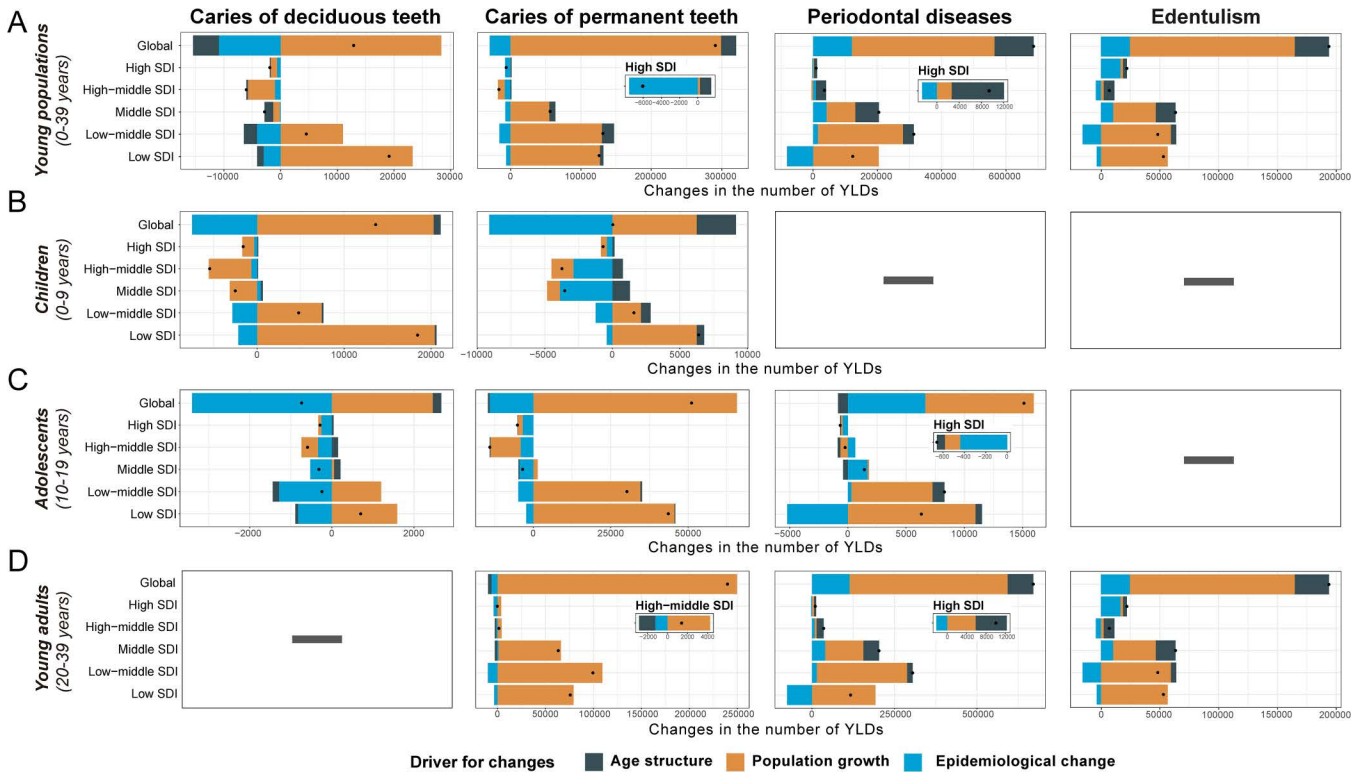

**Fig 4. Changes in the number of YLDs of oral disorders attributed to different drivers from 1990 to 2021 at the global level and by SDI quintiles. (A)** Changes in the number of YLDs for oral disorders among young population (0-39 years) attributed to different drivers from 1990 to 2021 at the global level and by SDI quintiles. **(B)** Changes in the number of YLDs for oral disorders among children (0-9 years) attributed to different drivers from 1990 to 2021 at the global level and by SDI quintiles. **(C)** Changes in the number of YLDs for oral disorders among adolescents (10-19 years) attributed to different drivers from 1990 to 2021 at the global level and by SDI quintiles. **(D)** Changes in the number of YLDs for oral disorders among young adults (20-39 years) attributed to different drivers from 1990 to 2021 at the global level and by SDI quintiles. YLD, year lived with disability; SDI, sociodemographic index; ASYLDR, age-standardized years lived with disability rate.

## 4. Discussion

This study comprehensively evaluates the burden of oral disorders among children, adolescents, and young adults (0–39 years) from 1990 to 2021, providing critical insights into lifecycle-specific challenges. Our findings align with previous GBD reports indicating a decrease in the YLD rate of caries and an increase in the rate of periodontal disease [3,4,6,10,13]. We extend this evidence by integrating age-stratified analyses with assessments of socioeconomic inequality, addressing the complex interplay of demographic and epidemiological factors across different developmental stages through subgroup analyses. The analytical approaches employed, including Joinpoint regression, decomposition analysis, and inequality analyses, collectively yield the study's key findings. For instance, Joinpoint regression identified 2015 as a turning point linked to United Nations Sustainable Development Goal 3.4 [29] initiatives and revealed the impacts of COVID-19 from 2019 to 2021. Decomposition analysis quantified how population growth outpaced epidemiological progress in low-SDI locations, while high-SDI locations effectively reduced both the number and rates of caries through successful preventive programs amidst population decline. Inequality analyses revealed persistent socioeconomic gradients in periodontal disease, even as absolute disparities narrowed. This integrated methodology avoids siloed interpretations, offering a cohesive narrative on oral health disparities.

The age range of 0–39 in this study encompasses critical transitions in oral health, from the development of primary dentition to the early signs of periodontal disease and edentulism in young adults. This age subgroup stratification aligns

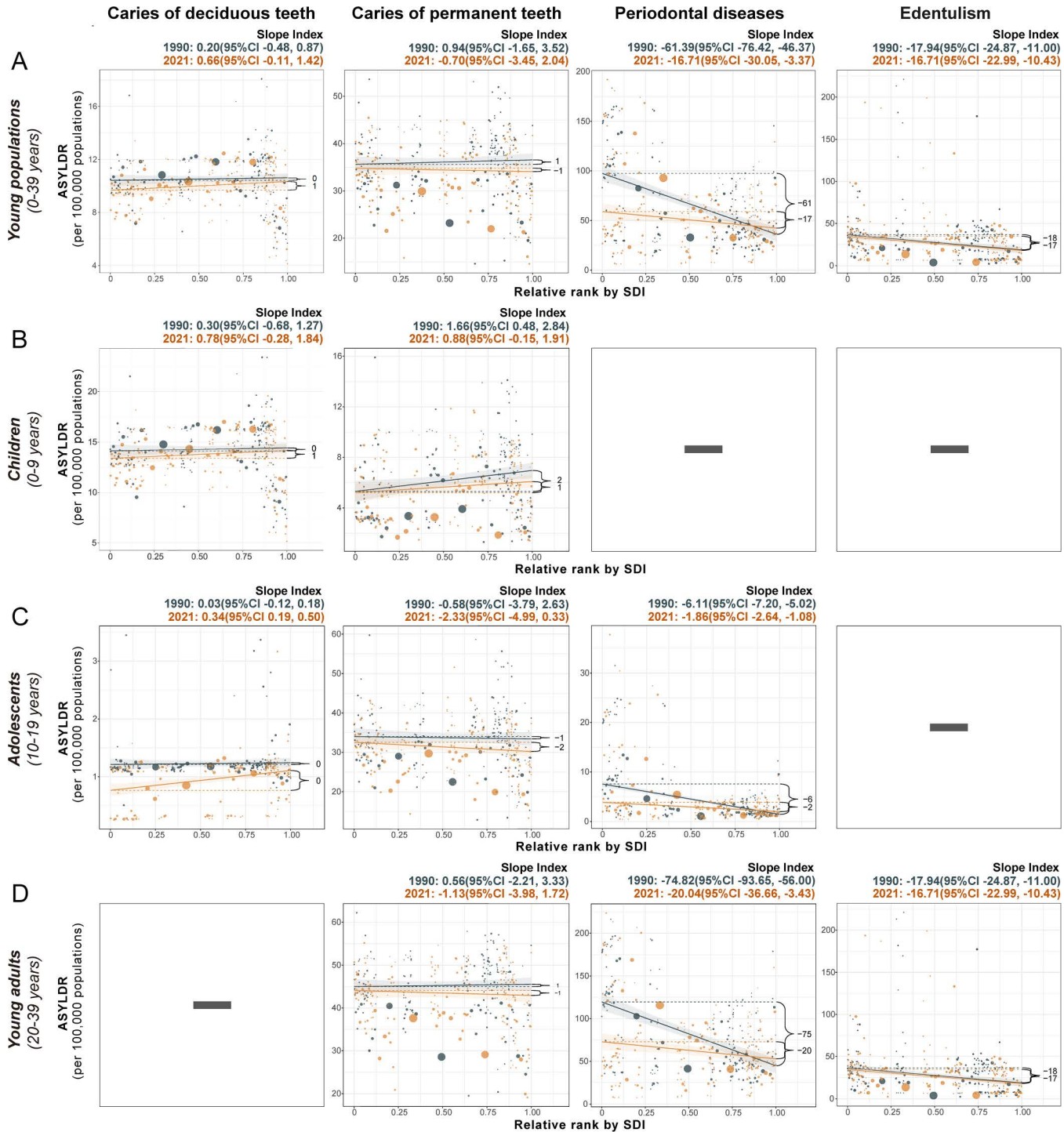

**Fig 5. SDI-related health inequality regression lines indicating absolute inequality in the ASYLDR of oral disorders, 1990 and 2021. (A)** SDI-related health inequality regression lines indicating absolute inequality in the ASYLDR of oral disorders among the young population (0-39 years). **(B)** SDI-related health inequality regression lines indicating absolute inequality in the ASYLDR of oral disorders among children (0-9 years). **(C)** SDI-related health inequality regression lines indicating absolute inequality in the ASYLDR of oral disorders among adolescents (10-19 years). **(D)** SDI-related health inequality regression lines indicating absolute inequality in the ASYLDR of oral disorders among young adults (20-39 years). Dots represent countries and territories; dot size represents population. CI, confidence interval; SDI, sociodemographic index; ASYLDR, age-standardized years lived with disability rate.

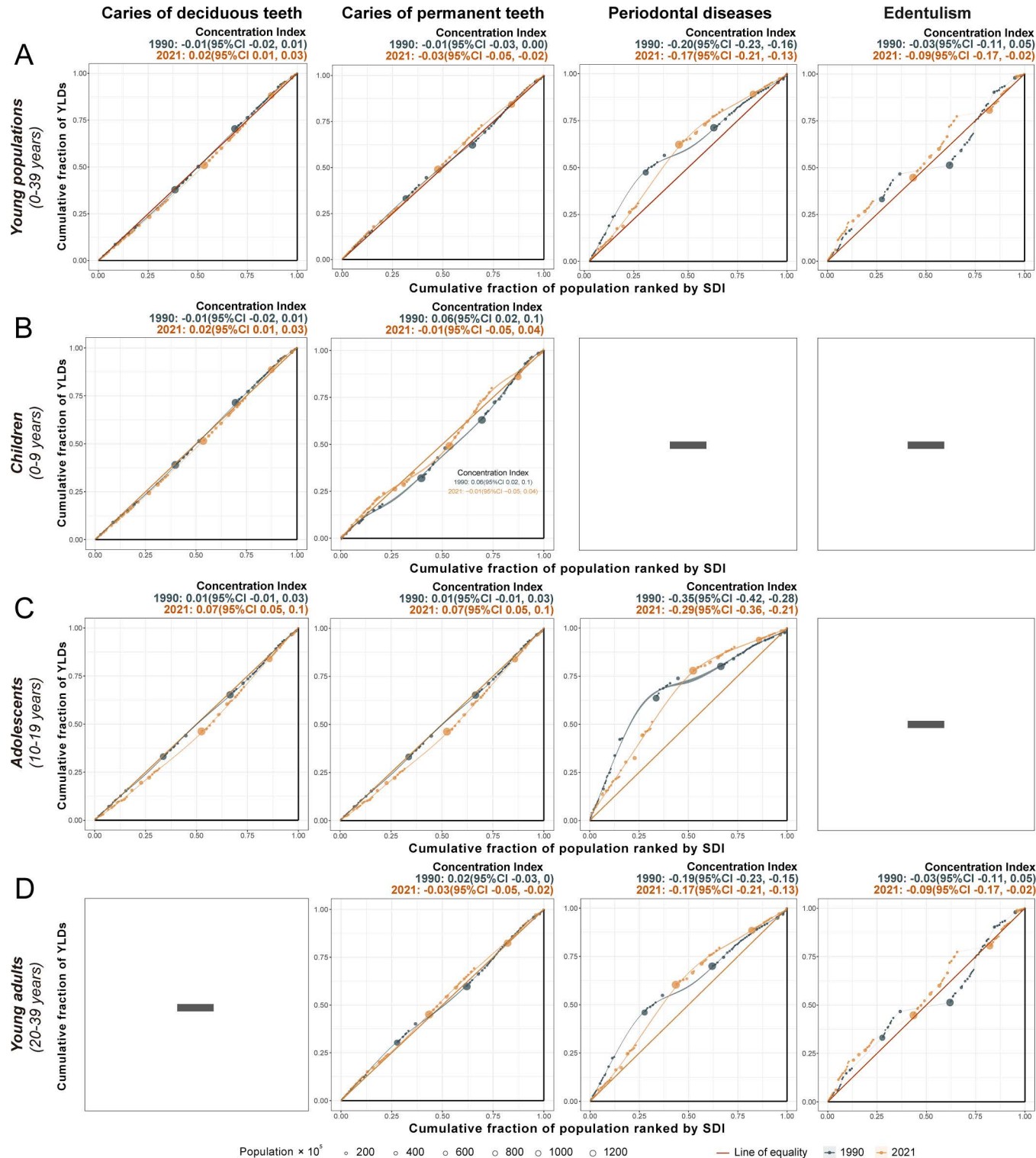

**Fig 6. SDI-related health inequality concentration curves indicating relative inequality in the ASYLDR of oral disorders, 1990 and 2021. (A)** SDI-related health inequality concentration curves indicating relative inequality in the ASYLDR of oral disorders among young population (0-39 years). **(B)** SDI-related health inequality concentration curves indicating relative inequality in the ASYLDR of oral disorders among children (0-9 years). **(C)**

SDI-related health inequality concentration curves indicating relative inequality in the ASYLDR of oral disorders among adolescents (10-19 years). **(D)** SDI-related health inequality concentration curves indicating relative inequality in the ASYLDR of oral disorders among young adults (20-39 years). Dots represent countries and territories; dot size represents population. CI, confidence interval; SDI, sociodemographic index; ASYLDR, age-standardized years lived with disability rate.

with the WHO's lifecycle framework [30] and previous GBD studies [31,32]. Existing evidence indicated that deciduous caries peaks between the ages of 5–9 [2] and juvenile (aggressive) periodontitis peaks at ages 15–19 within the 0–39 age range [33], further supporting our stratification. Attention to young edentulous adults is essential, as this condition is often associated with significant impacts on both their quality of life and psychological well-being [34]. The stratified lifecycle-specific insights address gaps in existing studies that have generalized across age groups [7,13]. By prioritizing YLDs over incidence and prevalence, the study emphasizes the prolonged disability and societal impact of untreated conditions. For instance, while the age-standardized prevalence rate of periodontal disease in young adults was lower than that of permanent caries, its higher ASYLDR reflects chronic disability and unmet treatment needs, necessitating policy shifts towards periodontal prevention and control strategies [6,12].

The observed epidemiological improvements in untreated caries over three decades, particularly among children and adolescents, underscore the success of preventive measures such as fluoride programs and school-based interventions. However, the delayed peak of untreated permanent caries burden into late adolescence and young adulthood highlights gaps in sustaining these gains. Notably, 2015 marked a key turning point in oral disorder trends, coinciding with global efforts to reduce non-communicable disease burdens as outlined in the United Nations Sustainable Development Goal 3.4 [29]. This led to decreased prevalence and YLDs of permanent caries, periodontal diseases, and edentulism in adolescents and young adults. However, the post-2015 rise in the prevalence and YLD rates of children's deciduous caries, exacerbated by COVID-19 disruptions, revealed vulnerabilities in childhood care. The pandemic and city lockdowns disproportionately affected children, whose limited dietary autonomy and prioritization of resources during emergencies increased sugar exposure [35–37]. Stress, routine changes, and limited access to healthier options may also have increased sugar consumption and caries progression, particularly in financially strained or less developed locations [38–40]. Moreover, reductions in school-based dental programs have diminished caries intervention for children [41,42]. Therefore, COVID-19 complicates the epidemiological trends of caries, necessitating age-targeted post-pandemic interventions.

Decomposition analyses revealed population growth as the dominant driver of burden increases in low-SDI locations, particularly among young adults. This cohort experienced the largest YLD surges in permanent caries and periodontal disease, attributable to demographic expansion in low-income settings, heightened demand for limited healthcare resources, and cumulative risk exposure (e.g., delayed care due to out-of-pocket costs). Conversely, in high-SDI locations, effective healthcare programs and population decline collaboratively facilitated reductions in untreated caries burdens among children and adolescents. Cross-country inequality analyses confirmed that periodontal disease disproportionately burdens low-SDI locations. Poverty, limited education, and poor living conditions negatively impact oral health outcomes, compounded by a lack of access to nutritious food, clean water, and healthcare services [8,43]. While SDI-related inequality of periodontal diseases improved from 1990 to 2021, absolute gaps remain entrenched, particularly among young adults. The trend towards promoting universal health coverage, including oral health care, has gained traction recently, but implementation varies widely among locations and countries [44]. This study advocates for the integration of oral health into universal health coverage, prioritizing early detection, treatment, and management of caries and periodontal disease in low-resource settings among vulnerable younger populations [45,46]. Collaborative efforts involving policymakers and healthcare providers are crucial to tailor universal health coverage strategies to address oral health needs.

Furthermore, integrating oral health indicators into public health monitoring systems is essential for tracking the distribution and trends of oral health conditions and developing targeted interventions [47]. Our analysis presents the latest distribution, trends, drivers, and inequality status of oral disorders in children, adolescents, and young adults, serving as

a foundational resource for future health policy development, particularly regarding interventions for vulnerable youth. However, limitations persist. Firstly, current GBD data only cover basic disorders (caries, periodontal diseases, edentulism), while other dental, tongue, and jaw disorders are grouped under 'other oral disorders', with data sourced solely from a nationally representative survey conducted annually from 1996 to 2011 by the US Agency for Healthcare Research and Quality, making accurate global interpretation impractical. Additionally, oral cancers were excluded from our analysis due to differences in burden presentation and risk factor attribution compared to non-fatal oral disorders. This aspect will be covered in a separate study. Moreover, the burden of oral disorders was influenced by detection methods, screening quality, and data recording, all linked to socioeconomic status. Disparities in oral healthcare services among countries suggest that burdens may be underestimated in nations with lower sociodemographic indices. The cross-country analysis of inequalities in oral disorder burdens does not aim for causal inference but serves as a tool for identifying patterns of inequality to guide policymaking. Therefore, the results warrant cautious interpretation. Future research should focus on large-scale real-world studies to validate our findings, emphasizing multivariate subnational analyses in oral disorder burdens.

## 5. Conclusion

This study reveals shifting patterns in the global burden of oral disorders among children, adolescents, and young adults from 1990 to 2021. Age-standardized YLD rates of caries have declined, while those of periodontal disease and edentulism have risen in all subgroups. Although epidemiological improvements have generally reduced the caries YLD rate in the past 30 years, especially in higher SDI locations, population growth in lower SDI locations has led to absolute increases in the burden of oral disorders globally. Number and age-standardized YLD rates of periodontal disease and edentulism have risen among adolescents and young adults, exacerbated by worsened epidemiology and population growth. Despite improvements in 2021 compared to 1990, inequalities in the burden of periodontal diseases persist, disproportionately affecting lower SDI locations. Continued investment in collaborative efforts for analyzing the burden changes and implementing preventive interventions is essential to address the disease burdens induced by population growth, especially in less-developed locations.

## Supporting information

**S1 Text.  Supplementary Methods.**
(DOCX)

**S2 Text.  GATHER Checklist.**
(DOCX)

**S3 Text.  Supplementary Figures and Tables.**
(DOCX)

**S4 Text.  Supplementary Results.**
(DOCX)

## Acknowledgments

We thank the original contributors of the GBD 2021 datasets systematically re-analyzed in this current study.

## Author contributions

**Conceptualization:** Yue Chen, Zhishen Jiang, Liu Liu, Hu Long.

**Data curation:** Jian Pan, Yubin Cao, Wenli Lai.

**Formal analysis:** Yue Chen, Zhishen Jiang.

**Methodology:** Yue Chen, Zhishen Jiang.

**Supervision:** Liu Liu, Jian Pan, Yubin Cao, Wenli Lai, Hu Long.

**Writing – original draft:** Yue Chen, Zhishen Jiang.

**Writing – review & editing:** Liu Liu, Jian Pan, Yubin Cao, Wenli Lai, Hu Long.

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
