## [Decision Letter · Decision Letter 0]

8 Sep 2025

PGPH-D-25-02129

Global and Regional Burden and Inequalities of Oral Conditions in Children, Adolescents, and Young Adults (0-39 years), 1990 to 2021

Dear Dr. Liu,

Thank you for submitting your manuscript to PLOS Global Public Health. After careful consideration, we feel that it has merit but does not fully meet PLOS Global Public Health’s publication criteria as it currently stands. Therefore, we invite you to submit a revised version of the manuscript that addresses the points raised during the review process.

We look forward to receiving your revised manuscript.

Kind regards,

Nicola Hawley

Academic Editor

Journal Requirements:

1. We have amended your Competing Interest statement to comply with journal style. We kindly ask that you double check the statement and let us know if anything is incorrect.

2. Some material included in your submission may be copyrighted. According to PLOS’s copyright policy, authors who use figures or other material (e.g., graphics, clipart, maps) from another author or copyright holder must demonstrate or obtain permission to publish this material under the Creative Commons Attribution 4.0 International (CC BY 4.0) License used by PLOS journals. Please closely review the details of PLOS’s copyright requirements here: PLOS Licenses and Copyright. If you need to request permissions from a copyright holder, you may use PLOS's Copyright Content Permission form.

Potential Copyright Issues: Figure 2: please (a) provide a direct link to the base layer of the map (i.e., the country or region border shape) and ensure this is also included in the figure legend; and (b) provide a link to the terms of use / license information for the base layer image or shapefile. We cannot publish proprietary or copyrighted maps (e.g. Google Maps, Mapquest) and the terms of use for your map base layer must be compatible with our CC-BY 4.0 license.

Additional Editor Comments (if provided):

Reviewer #1:

Reviewer #2:

Reviewers' comments:

Reviewer's Responses to Questions

**Comments to the Author**

1. Does this manuscript meet PLOS Global Public Health’s publication criteria?

Reviewer #1: Yes

Reviewer #2: Yes

2. Has the statistical analysis been performed appropriately and rigorously?

Reviewer #1: Yes

Reviewer #2: Yes

3. Have the authors made all data underlying the findings in their manuscript fully available (please refer to the Data Availability Statement at the start of the manuscript PDF file)?

Reviewer #1: Yes

Reviewer #2: Yes

4. Is the manuscript presented in an intelligible fashion and written in standard English?

Reviewer #1: Yes

Reviewer #2: Yes

Reviewer #1: This manuscript is scientifically rigorous, fills a critical evidence gap in youth oral health burden research, and provides actionable insights for equitable public health interventions. No revisions are needed—I strongly recommend direct acceptance for publication in PLOS Global Public Health.

Reviewer #2: Oral diseases are frequently overlooked when considering the global burden of noncommunicable disease, despite being a major public health concern. The authors of this manuscript, “Global and Regional Burden and Inequalities of Oral Conditions in Children, Adolescents, and Young Adults (0-39 years), 1990 to 2021,” utilize Global Burden of Diseases, Injuries, and Risk Factors Study (GBD) data to identify temporal trends (via joinpoint regression) and drivers (via decomposition analysis) of non-fatal oral diseases over three decades for young populations. Years lived with disability (YLDs), incidence, and prevalence for deciduous caries, permanent caries, periodontal disease, edentulism, and other oral disorders and sociodemographic index (SDI) quintile for countries and territories were obtained from GBD. Age-standardized rates were calculated for all phenotypes, and relative (concentration index) and absolute (slope index) inequalities in oral disease burden as well as the burden attributable to inequality in sociodemographic development were assessed. Periodontal disease was responsible for the most YLDs of the diseases present in the study, and population growth was identified as the major driver of YLDs globally. SDI-related inequalities were observed for periodontal disease only. A major strength of this study is the inclusion of stratified analyses for age (children, 0-9 years; adolescents, 10-19 years; young adults, 20-39 years) and SDI quintile (low, low-middle, middle, high-middle, high), which highlighted differences in the burden of disease and their temporal trends across these demographics. This study is helpful in explaining the global landscape of oral diseases over the last 30 years and illustrates the impacts of COVID-19 on oral health at a population level, particularly for children aged 0-9 years. These findings will inform public health strategies for oral disease prevention.

Major comments: None

Minor comments:

1. A maximum of 5 joinpoints was allowed for in the joinpoint regression, but no justification is provided for this parameter.

2. Table 1 is impossibly small and unreadable.

3. Figure 2 is fantastic, but the legends correspond to different age-standardized YLD rates (ASYLDRs), making comparisons across different age groups and among the various oral diseases challenging. Consider using the same scale for all plots in this figure.

**Do you want your identity to be public for this peer review?** For information about this choice, including consent withdrawal, please see our Privacy Policy

Reviewer #1: No

Reviewer #2: No

---

## [Editor Report · Decision Letter 1]

16 Sep 2025

Global and Regional Burden and Inequalities of Oral Conditions in Children, Adolescents, and Young Adults (0-39 years), 1990 to 2021

PGPH-D-25-02129R1

Dear Dr. Liu,

We are pleased to inform you that your manuscript 'Global and Regional Burden and Inequalities of Oral Conditions in Children, Adolescents, and Young Adults (0-39 years), 1990 to 2021' has been provisionally accepted for publication in PLOS Global Public Health.

Best regards,

Nicola Hawley

Academic Editor